

# Progress in the mechanism of autophagy and traditional Chinese medicine herb involved in alcohol-related liver disease

Wenwen Han[1,2,*], Haiyu Li[1,2,*], Hanqi Jiang[2], Hang Xu[2], Yifeng Lin[2], Jiahuan Chen[1], Chenchen Bi[2] and Zheng Liu[3]

[1] Department of Medical Laboratory, School of Medicine, Shaoxing University, Shaoxing, Zhejiang Province, China

[2] Department of Clinical Medicine, School of Medicine, Shaoxing University, Shaoxing, Zhejiang Province, China

[3] Department of Pharmacology, School of Medicine, Shaoxing University, Shaoxing, Zhejiang Province, China

[*] These authors contributed equally to this work.

Corresponding authors
Haiyu Li, lihy245@163.com
Zheng Liu, liuzheng1707@163.com

## ABSTRACT

Alcohol-related liver disease (ALD) is chronic liver damage caused by long-term heavy drinking with, extremely complicated pathogenesis. The current studies speculated that excessive alcohol and its metabolites are the major causes of liver cell toxicity. Autophagy is evolutionarily conserved in eukaryotes and aggravates alcoholic liver damage, through various mechanisms, such as cellular oxidative stress, inflammation, mitochondrial damage and lipid metabolism disorders. Therefore, autophagy plays an critical role in the occurrence and development of ALD. Some studies have shown that traditional Chinese medicine extracts improve the histological characteristics of ALD, as reflected in the improvement of oxidative stress and lipid droplet clearance, which might be achieved by inducing autophagy. This article reviews the mechanisms of quercetin, baicalin, glycycoumarin, salvianolic acid A, resveratrol, ginsenoside rg1, and dihydromyricetin inducing autophagy and their participation in the inhibition of ALD. The regulation of autophagy in ALD by these traditional Chinese medicine extracts provides novel ideas for the treatment of the disease; however, its molecular mechanism needs to be elucidated further.

## INTRODUCTION

Alcohol-related liver disease(ALD) consists of a series of liver diseases caused by excessive alcohol intake. It is characterized by metabolic changes in the liver, mitochondrial damage, destruction of lipid homeostasis, oxidative stress, and cell death caused by excessive drinking (*Rehm & Imtiaz, 2016*). With the widespread use of alcohol worldwide, excessive drinking has become the primary cause of chronic liver diseases, and alcohol consumption underlies approximately 50 of the deaths attributed to this disease (*Rehm & Shield, 2019*). Therefore, ALD has an urgent requirement for effective interventions.

The occurrence and development of ALD are associated with the increase in the number of lipid droplets (LDs) in liver cells caused by alcohol misuse and the oxidative damage of liver cells caused by alcohol metabolism, which together promote liver inflammation and subsequent hepatic fibrosis (*Michalak, Lach & Cichoz-Lach, 2021*). This phenomenon is also reflected in the four histological stages of ALD, alcoholic steatohepatitis (ASH), alcoholic hepatitis (AH), chronic hepatitis, and cirrhosis, which eventually lead to hepatocellular carcinoma (HCC) (*Ceni, Mello & Galli, 2014*). Although several studies have explored the pathophysiological mechanism of ALD, one of the major causes of liver diseases worldwide, the treatment of the disease is yet to be determined.

Due to the insidious and long-term nature of ALD, Western medicine often shows little curative effect. In the clinic, doctors prefer to prescribe traditional Chinese medicine against ALD. Traditional Chinese herbal extracts have been used to improve the liver tissue injury caused by alcohol and prevent ALD from developing into cirrhosis or liver cancer (*Yunhe et al., 2023*). Given their beneficial effects, traditional Chinese medicines are the "first choice" for maintaining health, but their precise role and mechanisms for regulating hepatocellular function remain unknown. In recent years, autophagy has been strongly linked to functional regulation of traditional Chinese herbal extracts in liver metabolism (*Li et al., 2023*).

Autophagy is a lysosomal degradation pathway evolutionarily conserved across eukaryotic cells. Under physiological or pathological conditions, eukaryotic cells can eliminate aging, damage, and excess organelles and contents through autophagy degradation that achieves material circulation and energy supply to maintain the homeostasis of the cell environment (*Giansanti, Torriglia & Scovassi, 2011*). Autophagy is up-regulated when subjected to extracellular or intracellular stress, starvation, growth factor deprivation, endoplasmic reticulum stress, and pathogen infection (*Rabinowitz & White, 2010*). Typically, the moderate activation of autophagy maintains cell and body homeostasis and ensures normal function. However, the overactivation of autophagy can also lead to cell death, and several diseases are related to autophagy. In liver diseases, accumulating evidence showed that autophagy is affected during the progress of ALD, leading to the accumulation of harmful metabolites of alcohol; however, the underlying mechanism is not yet clarified. These findings indicated that maintaining the appropriate degree of autophagy in liver cells is crucial.

Autophagy can be categorized into two types depending on whether the phagocytic object is selective. Selective autophagy is further subdivided into mitophagy, lipid autophagy, and ribosomal autophagy according to the phagocytic object (*Fimia, Kroemer & Piacentini, 2013*).

Autophagy is a complex dynamic process. Several positive and negative steps, including initiation, nucleation, extension, occlusion, and degradation, regulate the process. The molecular details of these steps have been elucidated. The autophagy process requires autophagy-related genes and also various protein complexes. To date, 30 autophagy-related genes and some regulatory proteins have been shown to be involved in autophagy (*Williams & Ding, 2020*). Firstly, the initiation of the process requires two proteins, the homolog ULK1 (Unc-51-like kinase 1) of atg1 and the homolog FIP200 of atg17, to

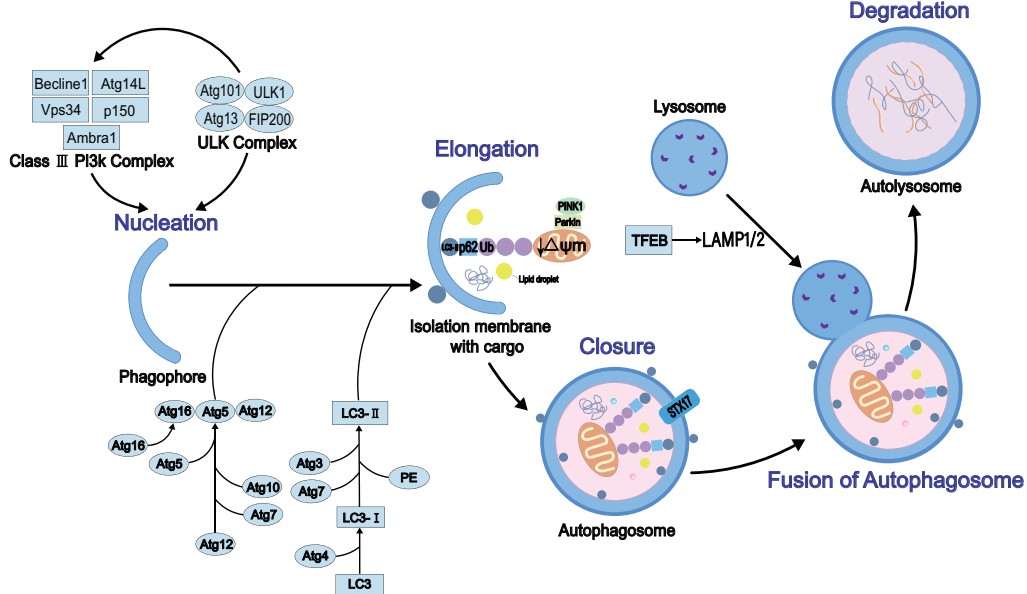

**Figure 1** **The general process of autophagy and its regulation in cells.** Physiologically, autophagy involves five key events: nucleation, elongation, closure, fusion, and degradation. In this process, several key proteins participate in its development and regulation, such as ULK protein kinase complex, mammalian target of rapamycin (mTOR), LC3, STX17 and AMPK.

form the ULK protein kinase complex that interacts with atg13 (*Mizushima, 2010*). The complex is negatively regulated by the nutrient sensor, termed rapamycin complex 1 (mTORC1), and the energy sensor, termed AMP-activated protein kinase (AMPK) (*Kim et al., 2011*). Then, vacuolar protein sorting 34 (Vps34), vacuolar protein sorting 15 (Vps15), Beclin-1 (homologous to Atg6), and Atg14L participate in the nucleation reaction of membrane vesicles to form preautophagosome (*Russell et al., 2013*). This process also involves two ubiquitin-like coupling systems, Atg7 (E1 ubiquitin-activating-like enzyme)-Atg3 (E2 ubiquitin-coupled-like enzyme)-microtubule-associated light chain (LC3) and Atg12-Atg5-Atg16L1 complex (E3 ubiquitin ligation-like enzyme), which regulate the coupling of phosphatidylethanolamine (PE) and LC3 (called LC3-II) (*Li & Ding, 2017*), essential for the autophagosome membrane extension and sealing step (*Li et al., 2021*). The sealing facilitates the fusion of autophagosome membranes fuse each other into enclosed bimodal vesicles. However, the specific mechanism is yet to be explored. Finally, autophagosomes and lysosomes are fused to form autophagolysosomes mediated by Ras-related protein 7 (Rab7) (*Kuchitsu & Fukuda, 2018*), lysosome-associated membrane protein1/2 (LAMP1/2) (*Kim, Choi & Kang, 2020*), and Syntaxin17 (STX17) (*Itakura, Kishi-Itakura & Mizushima, 2012*) that further regulate mediate the degradation of the cargo. The schematic of autophagy with key regulatory proteins is illustrated in Fig. 1.

## SURVEY METHODOLOGY

Data were searched from the PubMed (https://pubmed.ncbi.nlm.nih.gov/), Web of Science (https://www.webofscience.com), CNKI (https://www.cnki.net), GoogleScholar (https://scholar.google.com) and Wanfang (https://www.wanfangdata.com.cn) databases. The keywords used were as followes: autophagy, mitophagy, lipophagy, alcohol-related liver disease, alcoholic liver disease, Chinese herbal extracts, quercetin, baicalin, glycycoumarin, salvianolic acid A, Resveratrol, Ginsenoside Rg1, dihydromyricetin in both English and Chinese. We removed duplicate content, evaluated literature availability and selected the most recently published literature with a high impact factor. Studies that were not focused on the therapeutic effect of Chinese herbal extracts on ALD were excluded. We finally selected 103 articles for review.

## ALD AND AUTOPHAGY

The understanding of the complex and diverse pathogenesis of ALD is crucial to finding an effective treatment for ALD. An in-depth investigating of autophagy would clarify its role in ALD. The basic pathogenesis of ALD can be elaborated based on the liver cytotoxicity caused by alcohol metabolism (*Seth et al., 2011*). Most of the alcohol is metabolized in the liver. Under normal conditions, most of the alcohol consumed by the human body is oxidized to acetaldehyde by alcohol dehydrogenase (ADH) in the liver, and the acetaldehyde is oxidized into acetic acid by acetaldehyde dehydrogenase 2 (ALDH2); the acetic acid enters the tricarboxylic acid cycle and is metabolized to the final product. During the production of acetaldehyde, a part of NAD+ is concomitantly reduced to NADH (*Quintanilla et al., 2007*). Hence, during a high intake of alcohol, excessive acetaldehyde is converted into NADH, and an increase in NADH/NAD content in the cytoplasm, causes further damage to the liver cells. The mechanism underlying this phenomenon could be related to the removal of abnormal acetaldehyde due to the lack of ALDH2. In addition, alcohol metabolism leads to the accumulation of reactive oxygen species (ROS) (*Muñiz Hernández, 2012*). These effects of the products of excessive alcohol metabolism lead to protein aggregation, oxidative stress, mitochondrial damage, steatosis, and hepatocyte death, which prompts further exploration of ALD. Notably, ethanol inhibits the secretion of very low-density lipoprotein (VLDL), reduces the production of triglyceride (TG), and then aggravates the accumulation of liver lipids (*Van de Wiel, 2012*). In addition, the non-oxidative pathway of ethanol produces fatty acid ethyl ester (FAEE) and causes mitochondrial damage (*Heier et al., 2016*).

Several studies have shown that autophagy induced by alcohol stress protects the liver from alcohol-induced injury. Although yet to be elucidated, the mechanism may involve the selective degradation of damaged mitochondria or lipids (*Ding, Li & Yin, 2011*). Therefore, removing damaged mitochondria and excess LDs is the key to alleviating ALD. Moderate autophagy, including mitophagy and lipid autophagy, helps to recover liver function from metabolic disorders caused by alcohol that leads to damaged mitochondria, accumulation of fat and alcohol metabolites, ROS accumulation, *etc.*

Mitophagy is an autophagic process that specifically degrades damaged or redundant mitochondria. It is mediated by autophagy receptor proteins and can combine with damaged mitochondria (usually ubiquitinated) and LC3-modified phagocytic molecules (*Nguyen, Padman & Lazarou, 2016*). Several signaling pathways, including PINK1-Parkin signaling, are involved in mitochondrial degradation (*Geisler et al., 2010*). PINK1 is a mitochondrial serine/threonine kinase and is recruited to the mitochondria immediately after synthesis. It is cleaved by the presenilin-associated rhomboi-like protease (Parl) on the inner membrane and then rapidly degraded by the ubiquitin-proteasome pathway. When the mitochondria are damaged, mitochondrial membrane potential ($\psi$m) is reduced. Also, PINK1 is not cleaved and degraded, which leads to its accumulation in the outer membrane of the mitochondria, followed by its autophosphorylation, while the cytoplasmic Parkin is recruited to the organelle. Next, the activities of phosphorylated PINK1 and Parkin's E3 ubiquitin ligase are increased. The autophagosomes are then recruited to the mitochondria through receptor proteins, mitochondria are degraded through a series of reactions (*Eiyama & Okamoto, 2015*). Parkin is essential for mitophagy *in vivo* (*Whitworth & Pallanck, 2017*) (Fig. 2). In addition to association with neurodegenerative diseases, such as Parkinson's disease (*Ahmed et al., 2021*), *Ding & Yin (2012)* detected high expression levels of Parkin in mouse liver. The study also found that Parkin in mitophagy induction is a protective mechanism against alcohol-induced liver injuries and that damaged mitochondria are cleared out through Parkin-mediated mitophagy (*Kawajiri et al., 2010*; *Williams & Ding, 2015*).

Lipid autophagy (lipophagy) is a type of selective autophagy that removes redundant LDs by sending the content of LDs to lysosomes for degradation by autophagosomes (*Cingolani & Czaja, 2016*). Unlike adipose tissue, the liver is considered the organ of lipid metabolism (*Sathyanarayan, Mashek & Mashek, 2017*), and is vital for lipophagy in preventing LD accumulation and maintaining lipid homeostasis. Some studies have shown that rapamycin activates autophagy and decreases hepatic TG levels (*Ding et al., 2010*). Similar observations have been made in high fat-diet (HFD)-induced non-alcoholic fatty liver disease (NAFLD) models (*Lin et al., 2013*; *Schulze et al., 2017*), indicating that lipid removal involves a common lipid swallowing mechanism. Together, these findings suggested that autophagy is a protective mechanism in ALD, and enhancing the process is a potential method for the prevention and treatment of the disease (Fig. 3).

## EFFECT OF CHINESE HERBAL EXTRACTS ON AUTOPHAGY IN ALD

Traditional Chinese herbal extracts have been used for thousands of years. Although there is no sufficient theoretical basis, Chinese herbal extracts have been used to treat many diseases. With the advancement in biochemistry and pharmacological technology, in-depth research on the extraction and analysis of Chinese herbal extracts components and the mechanism of action has been conducted. Recent studies have shown that the induction of autophagy by Chinese herbal extracts has a positive effect on the progression of ALD (Table 1).

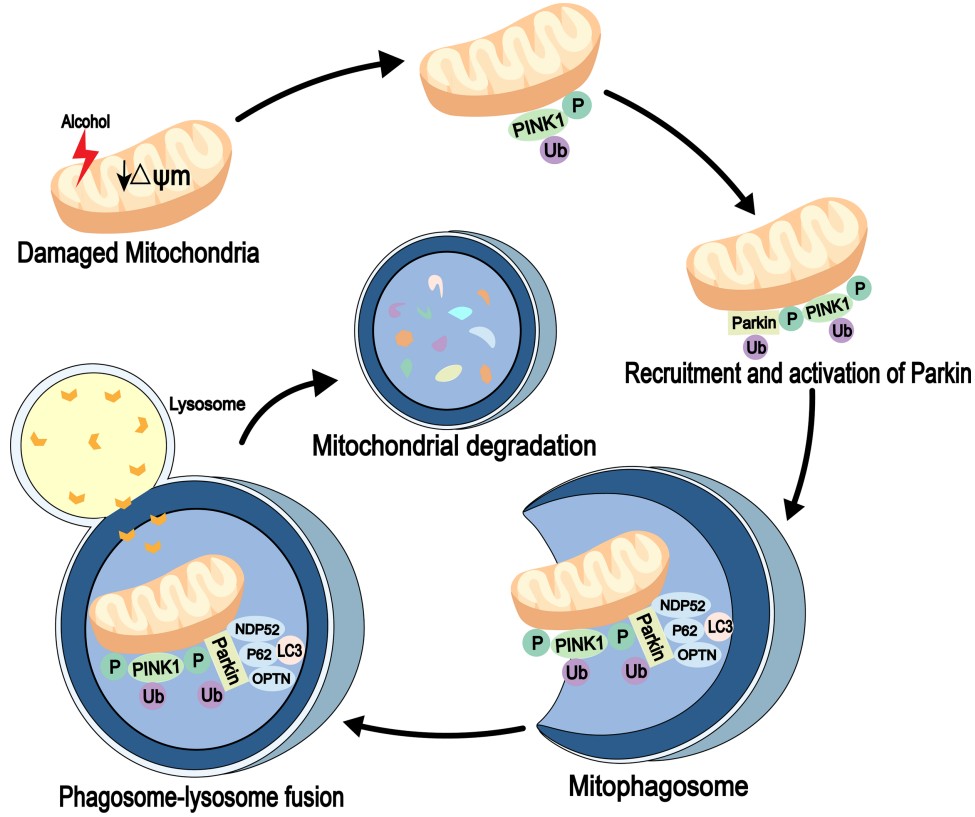

**Figure 2 Mitophagy pathway in alcohol injury.** PINK1-Parkin-dependent mitophagy involves the selective removal of damaged mitochondria through autophagy. When mitochondrial damage occurs, PINK1 accumulates on the outer mitochondrial membrane (OMM), leading to recruitment of Parkin through its phosphorylation function. Parkin then established polyubiquitin chains on the OMM to bridge the receptors p62, optineurin (OPTN) and nuclear domain 10 protein 52 (NDP52), and then connected the phagosome for mitochondrial isolation (mitophagosome). The mitophagosome then fuses with the lysosome to degrade the damaged mitochondria.

## Quercetin

Quercetin is a natural compound of multiple origins, found in foods, including vegetables, fruits, tea, and wine, as well as numerous food supplements (*Mlcek et al., 2016*). Because of its anti-inflammatory, anti-oxidation, antitumor, and other effects, quercetin has been widely used in cardiovascular disease, some forms of cancer, anti-aging, and other diseases (*Almatroodi et al., 2021*; *Mirsafaei et al., 2020*). Recently, quercetin has been identified as an effective liver protective agent.

The effect of quercetin on reducing liver LDs and improving liver histological abnormality has been observed in HFD-induced NAFLD (*Porras et al., 2017*) and diabetes mouse models (*Yang et al., 2019*). The hepatoprotective effect on NAFLD suggested that quercetin alleviates ALD. In the study of ethanol-induced liver injury rat-models, quercetin was reduces ALT/AST, shrinks the LDs, and restores the content of GSH and GPX. It also improves the oxidative stress and dysfunction of mitochondria, reduces the production of ROS and malonaldehyde (MDA) in liver mitochondria, and protects

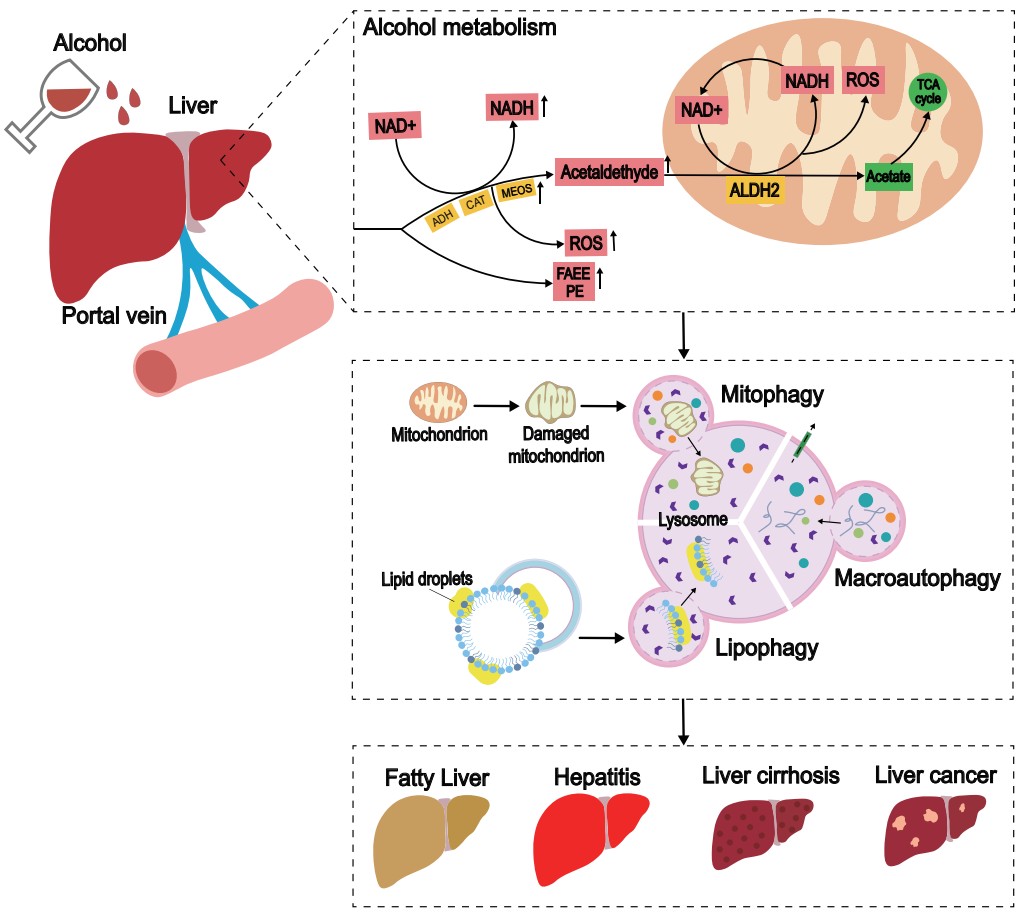

**Figure 3 The development of ALD caused by excessive alcohol intake.** In the liver, alcohol is metabolized through oxidative and non-oxidative pathways, with the oxidative pathway being the main pathway for alcohol metabolism. In the oxidative pathway, alcohol is oxidized by ADH to acetaldehyde, which is then decomposed to acetate. The non-oxidative pathway accounts for only a small amount of alcohol metabolism. Various enzymes non-oxidatively combine alcohol with different endogenous metabolites to produce FAEE and phosphatidylethanol (PEth). When excess alcohol is consumed, enzymes in the oxidative pathway are activated to promote the generation of ROS, and the substances produced by the non-oxidative pathway are also increased.

liver mitochondria from chronic alcoholism (*Tang et al., 2012*; *Zhao et al., 2022*). Another study showed that quercetin prevents ethanol-induced hepatotoxicity via p62-mediated atypical activation of the Nrf2 pathway (*Ji et al., 2015*). Coincidentally, the crosstalk between the Nrf2 pathway and autophagy indicated that activation of Nrf2 significantly alleviates ethanol-induced oxidative stress and subsequent liver damage (*Wang et al., 2015*). How Nrf2 regulates autophagy is not yet understood and may be related to p62. When autophagy is inhibited, accumulated p62 causes non-canonical activation of Nrf2, which in turn decreases autophagy activity (*Digaleh, Kiaei & Khodagholi, 2013*; *Zhou et al., 2013*). These findings indicated quercetin protects alcohol-induced liver damage. Some studies demonstrated that quercetin improves hepatic steatosis and liver injury of mice by chronic-plus-single-binge ethanol feeding, and the mechanism could be related to

**Table 1  Traditional Chinese herbal extracts for the treatment of ALD and the regulation of autophagic effects.**

| Active ingredient | Source | Impact on ALD | Model | Treatment | Regulatory target molecules in liver autophagy | Reference |
|---|---|---|---|---|---|---|
| | | | Animal model/Cell model | Animal model/Cell model | | |
| quercetin | Sophora flower-bud, vegetables, fruits | Improved the oxidative stress, lipid droplets, dysfunction of mitochondria, inflammation, steatosis | Cytotoxicity was induced by incubating L-02 cells (derived from an adult human normal liver) with ethanol (100 mM) for another 48h | Pre-incubated with quercetin (50 μM) for 15 min | P62-Nrf2 | Ji et al. (2015) |
| | | | Steatosis induced by chronic-plus-single-binge ethanol feeding (28% of total calories) for 12 wk in male C57BL/6J mice | Diet containing quercetin (100 mg/kg) for 12 wk | AMPK-mTOR | Zeng et al. (2019) |
| | | | ALD induced by chronic-plus-single-binge ethanol feeding (30% of total calories as ethanol) for 15 wk in male C57BL/6J mice | Diet containing quercetin (100 mg/kg bw) for 15 wk | mTOR-TFEB | Li et al. (2019) |
| | | | | Oral gavage feeding with quercetin (100 mg/kg) per day | AMPK-FOXO3a-Pink/Parkin | Yu et al. (2016) |
| baicalin | scutellaria baicalensis | Reduced lipid accumulation | ALD induced by 4 and 8 wk of 65% (vol/vol) alcohol intaking in Male Wistar rats | Oral feeding with baicalin concentration (120 mg/kg/day) for 4 wk | Bcl-2/Beclin 1 | Wang et al. (2016) |
| | | | ALD induced by 65% ethanol for 6 wk in C57BL/6 mice | Oral feeding with 50 mg/kg per day baicalin for 6 wk | NF-κ B-Beclin 1-P65 | Fang et al. (2022) |
| glycycoumarin | liquorice | Improved lipid metabolism, hepatotoxicity | Chronic model induced by Lieber-DeCarli ethanol diet for 5 wk and acute model induced by alcohol (6 g/kg bw) in C57BL/6J mice. | Diet containing GCM (10, 20 mg/kg) for 5 wk in chronic model and GCM (20 and 30 mg/kg. IP) for 3 days in acute model | P62-Nrf2 | Hassanein et al. (2020) and Song et al. (2015) |
| salvianolic acid A | salvia miltiorrhiza | Reduced lipid accumulation, steatosis, oxidative stress | ALD induced by LieberDeCarli ED diet for 8wk in Male SpragueDawley (SD, SCXK 2008-0002) rats | Diet containing different dioscin concentrations (8, 16/mg/kg/d) for 8 wk | AMPK-SIRT1 | Shi et al. (2018) |
| Resveratrol | Polygonum cuspidatum, mulberries, peanuts, grapes | Reduced lipid accumulation oxidative damage, cell death, mitochondrial damage | AFL induced by chronic-plus-single-binge ethanol Lieber–DeCarli liquid diet (67.3ml/L) for 15 days in male C57BL/6J mice | Oral gavage feeding with different RES doses (10 mg/kg, 30 mg/kg, 100 mg/kg) by gavage from day 10 to day 16 | mTOR | Tang et al. (2016) |
| | | | Steatosis was induced by incubating HepG2 cells with 100 μM oleic acid and 87 mM alcohol (OpA media) for 48 h | Treated with 45 μmol RES with or without 5 mmol/L 3-MA for 48h | | |
| Ginsenoside Rg1 | Ginsenoside | Improved lipid synthesis | ALD induced by Lieber-DeCarli ethanol liquid diet (36% ethanol) for 10 days in C57BL/6 male mice | Treated with Rg1 at various concentrations (10, 20 and 40 mg/kg/bw) for 10 days | AMPK | Gao et al. (2016) |
| dihydromyricetin | Grapes, vine tea | Improved liver inflammation and lipid oxid | ALD induced by Lieber-DeCarli liquid diet, which increasing concentrations of ethanol (1% for 2 d, 2% for 2 d, 4% for 7 d and then 4% for an additional 6 wk) | Oral feeding with two different doses (75 and 150 mg/kg BW per day) of DMY | Nrf2-P62/Keap-1 | Qiu et al. (2017) |
| | | | ALD induced by 30% EtOH single bottle access for 6 wk in male C57Bl/6J mice | Intraperitoneal injection of either 5 or 10 mg/kg of DHM throughout the entire study | AMPK | Silva et al. (2020) |

AMPK-mediated lipophagy. Typically, quercetin alleviates hepatic steatosis by reducing PIN2 levels, activating AMPK, and increasing the co-localization of liver LC3-II and PIN2 proteins to positively regulate autophagy (Zeng et al., 2019).

A mouse model study of ethanol-containing Lieber De Carli liquid diet showed that quercetin increases the expression of LAMP1, LAMP2, and Rab7 via nuclear translocation of transcription factor EB (TFEB) (Li et al., 2019), followed by the fusion of autophagosomes and lysosomes. This increased the autophagy that was manifested as the reduced

accumulation of abnormal LC3-II and p62. In addition, steatosis and inflammatory factors (IL-6 and TNF $\alpha$) were significantly down regulated, indicating that TFEB in the liver cell nucleus of mice fed alcohol for 15 weeks was substantially reduced compared to that in the matched control group. Moreover, mTOR and ERK2 are activated in the liver. However, quercetin inhibits mTOR activity, but no significant change was detected in the hyperphosphorylation of ERK2 induced by alcohol exposure. In summary, quercetin promotes the nuclear translocation of TFEB by inhibiting mTOR. Therefore, mTOR-TFEB can be used as a drug target to prevent alcoholic liver injury through autophagy, which needs to be investigated further.

Reportedly, quercetin reverses the inhibitory effect of ethanol on liver mitochondrial autophagy by enhancing Parkin-dependent mitophagy (*Yu et al., 2016*), which in turn combats alcohol-induced liver injury (*Williams & Ding, 2015*; *Williams et al., 2015*). Compounds reduce alcohol-induced liver injury by regulating mitophagy (*Zhao et al., 2021*; *Zhou et al., 2022*). These results suggested a protective role of mitophagy in alcohol-induced liver injury and that targeting it may be a prospective treatment for ALD.

Forkhead box O3 (FoxO3a) promotes autophagy to prevent ethanol-induced liver injury by activating the Pink/Parkin pathway (*Webb & Brunet, 2014*). Under chronic ethanol induction, FoxO3a is regulated by inhibited AMPK and ERK, and the expression and activity of FoxO3a are impaired (*Michalak, Lach & Cichoz-Lach, 2021*). Quercetin restores mitophagy by promoting the nuclear translocation of FoxO3a through AMPK and ERK2 pathways (*Yu et al., 2016*).

These findings suggested that quercetin prevents the development of ALD by regulating autophagy.

## Baicalin

Baicalin is extracted from the root of Scutellaria baicalensis Georgi. It reduces cerebrovascular resistance and is often used for the treatment of paralysis after cerebrovascular disease (*Li et al., 2020*). Moreover, due to its anti-inflammatory, antioxidant, and anti-apoptotic effects, baicalin has attracted significant attention as a hepatoprotective agent. Dai et al. found that chronic treatment with baicalin improves diet-induced obesity and hepatic steatosis by activating hepatic CPT1, which accelerates lipid oxidation (*Dai et al., 2018*).

In the treatment of ALD, administration of baicalin increased the expression of SOD and GSH-Px and reduced MDA production in the liver tissues compared to the control group. Also, the expression of SOD and GSH-Px increased in liver tissue, while the production of MDA decreased compared to the control group. This finding demonstrated that baicalin alleviates alcohol-induced oxidative stress in the liver.

Increased phosphorylation of Bcl-2 is considered to be anti-ALD based on the antioxidant properties of Nrf2 or Sonic hedgehog (Shh) pathway (*Wang et al., 2016*). Because the anti-apoptotic factor Bcl-2 binds to Beclin-1, when the Bcl-2 level is increased, the affinity between Bcl-2 and Beclin-1 is decreased and the autophagy flux is increased (*Fimia, Kroemer & Piacentini, 2013*).

Recently, *Fang et al. (2022)* showed that baicalin relieves ALD by increasing the expression of miR-205 to inhibit α5-mediated NF-κB signaling pathway. However, the mechanism of how importin α5 inhibits NF-κB remains to be elucidated. Some studies have shown that this phenomemon is related to the decline in p65, IκB α, and IKK β levels (*Fang et al., 2022*), which might also be noted when the inflammatory signaling pathway NF-κB is activated under the stimulation of excessive alcohol. Subsequently, Beclin-1 is released, IKK complex is activated, and IκB is phosphorylated and then ubiquitinated to release p50/p65 compounds (*Min et al., 2018*). The mutual regulation of NF-κB and autophagy is complex. NF-κB stimulates Beclin-1 to enhance autophagy, which in turn degrades IKK subunit and the NF-κB signaling pathway. The decreased expressions of IKK and IκB indicate the possibility of autophagy.

## Glycycoumarin

Licorice is a common edible and medicinal plant that has antioxidant, antigenotoxic, and anti-inflammatory activities (*Siracusa et al., 2011*). It can treat several diseases, including gastrointestinal disorders, respiratory infections, tremors (*Pastorino et al., 2018*), asthma with coughing, fatigue, debilitation (*Wang et al., 2013*), cancer, spasmodic diseases (*Song et al., 2015*), and liver diseases (*Chigurupati et al., 2016*; *Zhang et al., 2017*). Moreover, it does not work as a single agent but rather in combination with other medicines.

Hitherto, hundreds of bioactive components and plant metabolites of licorice have been extracted and isolated from Glycyrrhiza species (*Cevik, Kan & Kirmizibekmez, 2019*). Coumarin is one of the main bioactive chemical components in licorice (*Zhang et al., 2020*).

Glycycoumarin (GCM) is a representative of coumarin in licorice with, favorable bioavailability and multiple biological activities (*Qiao et al., 2014*). It can defend or treat miscellaneous liver diseases, including alcoholic liver injury (*Jung et al., 2016*), acetaminophen (APAP)-induced hepatotoxicity (*Yan et al., 2018*), and the disorder of lipid metabolism (*Zhang et al., 2017*). Some studies showed that the combination of GCM and alcohol greatly reduces the levels of ALT and AST in acute or chronic alcohol-induced liver injury (*Song et al., 2015*; *Zhang et al., 2020*), indicating that glycyrrhizin has a good hepatoprotective effect on reducing hepatotoxicity caused by alcohol exposure.

In addition, GCM inhibits hepatic steatosis in NASH mouse model and oleic acid/palmitate acid (OA/PA)-induced cell culture model by reducing lipid accumulation in the liver and blood. These experiments indicated that GCM regulates autophagy in hepatic steatosis through activating AMPK-mediated lipophagy pathway. After treatment with CGM, the phosphorylation of liver kinase B1(LKB1) the classical upstream kinase of AMPK, increased. Finally, autophagy was activated by increasing the phosphorylation of ULK1. Furthermore, AMPK was activated followed by inhibition of m-TOR, which in turn resulted in increased fatty acid b-oxidation and enhanced lipophagy (*Zhang et al., 2017*). Together, these findings suggested that GCM activates autophagy to protect the liver from lipid accumulation through the AMPK/mTOR pathway.

*In vivo* or *in vitro*, the levels of autophagy, including LC3-II, p62, and autophagic vacuoles (AVs), increased in the glycycoumarin treatment group compared to the

alcohol treatment group (*Song et al., 2015*), indicating that autophagy plays a role in the prevention of ALD. Another study showed that glycycoumarin improves ethanol-induced hepatotoxicity via Nrf2 activation (*Hassanein et al., 2020*) and autophagy. Nrf2 can be activated by glycycoumarin to combat oxidation and inflammation. Glycycoumarin up-regulates MAPKS, ERK1/2, and P38. After the inhibition of P38, Nrf2 and its two transcriptional targets HO-1 and GCLC were elevated, indicating that glycycoumarin improves alcohol-induced liver injury by increasing the levels and phosphorylation of P38 and Nrf2. Moreover, Nrf2 up-regulates of p62, which in turn inhibits Keap1. Then, it activates Nrf2 induced by glycycoumarin, forming a positive feedback loop of the molecule activation (*Song et al., 2015*).

Glycycoumarin and active ingredients exhibit various pharmacological activities (*Hassanein et al., 2020*) and are cost-effective. Notably, licorice and its active ingredients cause apparent mineralocorticoid excess (AME) syndrome and result in some side effects, such as inhibition of renin-angiotensin-aldosterone system, potassium loss, and sodium retention (*Olukoga & Donaldson, 2000*).

## Salvianolic acid A

Salvia miltiorrhiza, known as "Danshen," is a traditional and natural medicinal plant. In China, the extracts of several chemical constituents are derived from Salvia miltiorrhiza root. According to the chemical structure, these components can be divided into liposoluble tanshinones and water-soluble phenolics. The water-soluble phenolics are salvianolic acid A (SalA), salvianolic acid B (SalB), lithospermic acid, danshensu, caffeic acid, and rosmarinic acid (*Ren et al., 2019*). SalA and SalB are the main components of salvia miltiorrhiza that exert a protective role in the liver. SalB improves hepatic LD accumulation, SalA activates autophagy and improves liver steatosis, inflammation, and oxidative stress (*Ding et al., 2016*). In the study using chronic ethanol-induced liver injury models, Shi et al. confirmed that SalA reduces the level of ALT/AST and LDs (*Shi et al., 2018*; *Shi et al., 2018*), while the level of MDA decreases and the content of GSH and ROS-2 proteins increases. These findings indicated that SalA resists oxidative stress and mitochondrial dysfunction in the liver injury induced by ethanol (*Shi et al., 2018*). Songtao Li examined the effect of SalA on NAFLD models induced by a high-fat carbohydrate diet (HFCD). The results showed that SalA alleviates liver steatosis, liver injury and weight gain induced by HFCD via the AMPK/SIRT1 pathway. In addition, the expression of LC3-II was decreased, indicating that SalA alleviates NAFLD by activating the AMPK/SIRT1 pathway and autophagy (*Li et al., 2020*). Another study affirmed that SalA strengthens the activity of SIRT1 and promotes autophagosome-lysosome fusion to reverse the symptoms of ALD (*Shi et al., 2018*). The findings proved that the level of cathepsin B, LAMP2, and RAB7 was decreased in the ethanol group, which could be reversed by SalA, indicating that SalA restores the formation of lysosome and promotes the fusion of Av-lysosome. In conclusion, it could be put forth that SalA activates autophagy by upregulating SIRT1 and exerts liver-protective effects. Other studies have shown that SalA significantly improves liver function, liver morphology, and histology by inhibiting PI3K/AKT/mTOR and caspase-3 and activating

the Bcl-2 signaling pathways (*Wang et al., 2019*). Thus, these factors and pathways are deemed to be involved in autophagy.

## Resveratrol(RES)

RES(3,4′,5-trihy-droxystilbene) is a natural phytoalexin polyphenol that exists in more than 300 types of edible plants, such as grape, blueberry, cranberry, peanut, and cocoa. It is used to against trauma, bacteria, infection, UV, and other external environmental stresses (*Meng et al., 2021*).

Some studies have shown that RES improves alcoholic fatty liver (AFL) and reverses hepatitis by activating the mTOR and cAMP-PKA-AMPK pathways in autophagy. AFL is the early stage of ALD, and its treatment is crucial for the effective control of further deterioration of the condition. *Tang et al. (2016)* found that RES significantly reduces liver steatosis and inhibits the further development of ALD in the mouse model. The study also showed that RES reverses the high-density lipoprotein cholesterol level compared to the control group (*Tang et al., 2016*). Additionally, the results of hematoxylin and eosin (H&E) staining of the liver tissue showed that RES reduces liver fat accumulation but up-regulates the expression of LC3-II, Atg5, Beclin-1, and ULK1 , p62 and LC3-II/I ratio. These phenomena exacerbate the increase in the number of autophagosomes (*Ji et al., 2015*; *Milton-Laskibar et al., 2018*).

Taken together, it can be deduced that RES protects the liver and impedes ALD development by regulating the autophagy process.

In addition, some studies proved that RES affects lysosome acidification to activate and regulate autophagy in hepatocytes, which in turn improves the alcohol-induced damage to hepatocytes (*Tang et al., 2016*). Also, SIRT1 plays a role in autophagy, as shown by RES extract, because RES significantly regulates SIRT1 activity and increases autophagy in hepatocytes (*Zhang et al., 2015*).

## Ginsenoside

Panax ginseng is a perennial herb belonging to the genus Panax L. In modern medical theory, ginseng plays a regulatory role in many diseases (*Guo et al., 2021*), including chronic conditions, such as hypertension and diabetes mellitus (*Yuan et al., 2012*). As a natural plant supplement, ginseng drinks, ginseng sugar, ginseng tea, ginseng honey tablets, and other ginseng foods have gradually developed worldwide. In recent decades, with the development of extraction and separation technology, many active components have been identified in ginseng, along with their structure, activity, and function (*Zhou et al., 2019*).

### Ginsenoside Rg1

Rg1 has a various of pharmacological activities and also exerts a protective effect on the liver by regulating different pathways in several diseases (*Zhou et al., 2019*). During ALD, ethanol exposure decreases the mRNA expression of AMPK $\alpha$2, AMPK $\gamma$1, and PPAR $\alpha$, but Gao et al. showed that Rg1 treatment counteracts these changes and accelerates the rate of lipid synthesis and decomposition (*Gao et al., 2016*; *He & Klionsky, 2009*). Similarly, *Mao et al. (2016)* demonstrated that Rg1 decreases the expression of LC3-II and Beclin-1

and the formation of autophagosomes, *i.e.*, Rg1 reduces hepatic lipid accumulation and damage from lipid metabolism by regulating the AMPK pathway in autophagy.

### Dihydromyricetin

Dihydromyricetin is a dihydroflavonol compound found in plants of the grape family, vine tea family, Myriceaceae, and willow family. The content of dihydromyricetin in vine tea reaches up to 25%, which is higher than that in the stem (*Hou et al., 2015*). Also, the content of dihydromyricetin in vine tea varies across regions. As a flavonoid, dihydromyricetin has anti-inflammatory and antioxidant properties of general flavonoids, relieves alcohol poisoning, and prevents alcoholic liver, fatty liver, and other liver protective effects.

According to *Qiu et al. (2017)*, dihydromyricetin promotes the maintenance of GSH and the consumption of MDA in the liver, confirming that this drug prevents alcohol-induced liver injury. Mechanically, as an antioxidant, dihydromyricetin significantly induces autophagy by activating the Nrf2/Keap-1 pathway and enhances the degradation of Keap-1 and the abundance of Nrf2 by up-regulating the expression of p62, further promoting autophagy (*Qiu et al., 2017*).

Another study showed that dihydromyricetin improves liver injury and significantly reduces serum ALT and AST levels, the markers of liver injury, in alcohol-fed mice. Moreover, the inflammatory markers of liver injury, such as IFN-c and TNF- $\alpha$, are significantly decreased. In addition, the phosphorylation level of AMPK increases markedly, which inhibits FFA synthesis and increases the lipid oxidation of fatty acids to mitochondria. Also, the experimental data also showed that chronic alcohol feeding significantly increases Nrf2 (*Silva et al., 2020*). The increased expression of Nrf2 and AMPK promotes fat and alcohol metabolism and can also enhances autophagy.

## CONCLUSION

ALD is a general term used to refer to a broad spectrum of diseases, including asymptomatic early ALD (fatty liver or steatosis), steatohepatitis, advanced ALD (alcoholic hepatitis, cirrhosis), and HCC attributable to alcohol consumption. In ALD, alcohol consumption causes hepatic metabolic changes, oxidative stress, accumulation of lipid droplets, damaged mitochondria and cell death in hepatocytes. These changes and their abnormal synergy together promote the occurrence and progression of ALD (*Liu, Tsai & Hsu, 2021*).

At present, increasing evidence has shown that alcohol-induced abnormity can be regulated by autophagy (*Li, Ouyang & Zhu, 2023*). Autophagy is a critical self-protection and survival mechanism of eukaryotic cells that exerts a protective role in ALD by scavenging damaged mitochondria (*Thomes et al., 2015*), accumulated LDs (*Lin et al., 2013*; *Schulze et al., 2017*), and protein polymers (*Bonilla et al., 2013*). The underlying mechanisms in mammalian autophagy are mTOR (*Heras-Sandoval et al., 2014*) and AMPK pathways (*Mihaylova & Shaw, 2011*), endoplasmic reticulum stress, and ROS regulation (*Li et al., 2015*), and other autophagy regulatory molecules, such as FoxO3a (*Warr et al., 2013*) and TFEB (*Medina et al., 2015*). In this review, we summarized the process of autophagy and many biological targets that play an important role in autophagy, as well as their related signaling targets in alleviating ALD (Fig. 4).

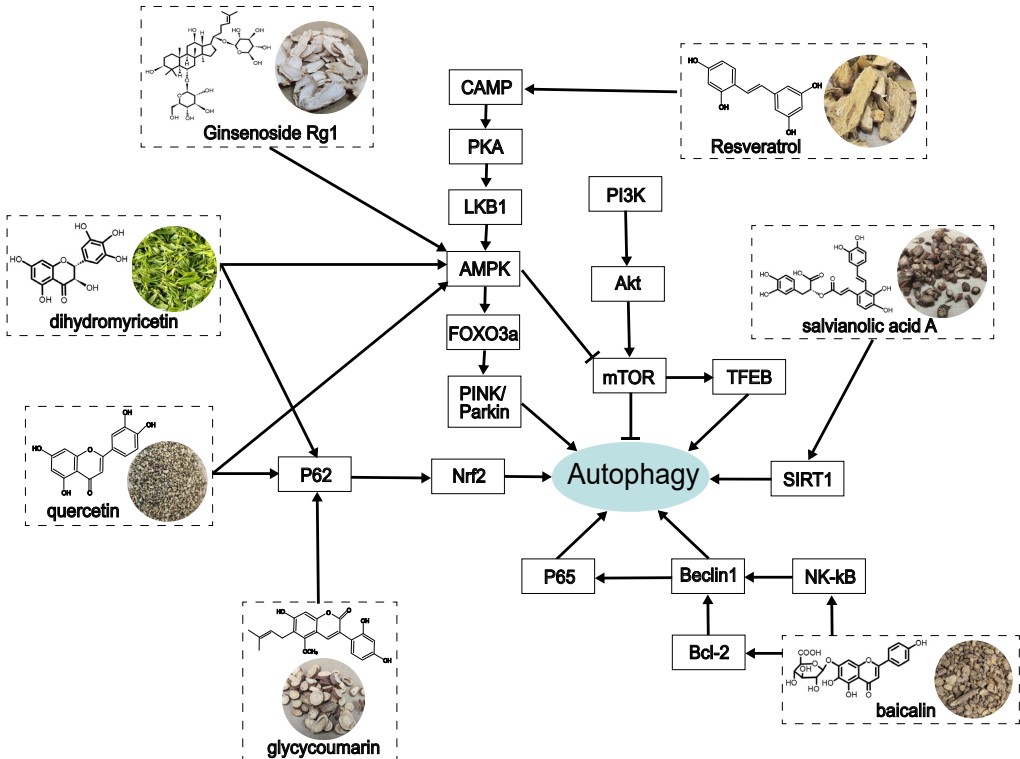

**Figure 4** **The role of traditional Chinese herbal extracts on autophagy in treating ALD.** Seven extracts modulated different aspects of autophagy by acting on the signaling of autophagy in the liver. PI3K, phosphoinositide 3-kinase; Akt/PKB, protein kinase B; mTOR, the mammalian target of rapamycin; CAMP, cyclic adenosine monophosphate; PKA, protein kinase A; LKB1, liver kinase B1; AMPK, AMP-activated protein kinase; FoxO3a, forkhead box O3a; PINK, PTEN-induced putative kinase; TFEB, transcription factor EB; SIRT1, Sirtuin1; Nrf2, nuclear factor erythroid 2-related factor 2; NF-κ B, nuclear factor kappa-B.

Traditional Chinese herbal extracts are therapeutic preparations in chronic and intractable diseases and have been applied to various diseases, including heart disease, stroke, flu, hepatopathy, cancer, and Alzheimer's disease (*Xiu et al., 2015*). Existing studies have shown that herbal extracts improve alcohol-induced liver tissue injury and prevent ALD from developing into cirrhosis or liver cancer.: Herein, we summarized the process of autophagy and many biological targets that play an important role in autophagy, as well as their related signaling targets in alleviating ALD. However, the detailed function and precise mechanism of many biological molecules involved in this process need to be further verified. For instance, some signaling molecules are yet to be elucidated, including Nrf2, LKB1, importin α5, and SIRT1 in cAMP-PKA-AMPK and PI3K/Akt pathways on alcoholic liver injury by regulating autophagy. Additionally, further attention should be paid to the study of the chemical structure of traditional Chinese medicine, in order to explore the therapeutic effects of the drugs on ALD mediated by autophagy. Because existing studies indicate that different herbal extracts have similar biological functions and similar molecular structures. For example, quercetin and ginseng are both flavonoids and

have similar effects, but the precise active ingredient is yet to be identified. This interesting finding also provide clues for improving the efficacy and selectivity by modifying the molecular structure. At the same time, new drugs that may have therapeutic effects on ALD can be found in the same type of compounds. Finally, there is a lack of comprehensive evidence on the dosage and safety evaluation of drugs in animal experiments related to the traditional Chinese medicine mentioned in this report (Table 1). This will seriously restrict the application of traditional Chinese medicine and should be included in future experimental design. Because research has shown that alcohol induced mouse models from different laboratories have bias in experimental results due to different designs. In future studies, the standardized ALD model and systemic evaluation criteria will be needed to increase the precision in evaluating autophagy regulation in the liver function of ALD.

In short, despite these shortcomings mentioned above, this study provides an insight into the molecular mechanism of alcohol metabolism and autophagy, suggesting the possible molecular mechanism of the hepatoprotective effect of these Chinese herbal extracts on ALD. Modulating autophagy by using Traditional Chinese medicine could provide novel therapeutic approaches for treating ALD. Certainly, further studies are required to analyze the role of these pathways in ALD and validate these findings in a clinical setting.

### Funding

The Natural Science Foundation of Zhejiang Province (No. LGF21H180005) and the Scientific Research Fund of Shaoxing University (Grant Nos. 202110349205xj) provided funding for the publication fee. The funders had no role in study design, data collection and analysis, decision to publish, or preparation of the manuscript.

### Grant Disclosures

The following grant information was disclosed by the authors:
The Natural Science Foundation of Zhejiang Province: LGF21H180005.
The Scientific Research Fund of Shaoxing University: 202110349205xj.

### Competing Interests

The authors declare that there are no competing interests.

### Author Contributions

- Wenwen Han conceived and designed the experiments, performed the experiments, analyzed the data, prepared figures and/or tables, authored or reviewed drafts of the article, and approved the final draft.
- Haiyu Li conceived and designed the experiments, performed the experiments, analyzed the data, prepared figures and/or tables, authored or reviewed drafts of the article, and approved the final draft.
- Hanqi Jiang conceived and designed the experiments, analyzed the data, prepared figures and/or tables, and approved the final draft.

- Hang Xu conceived and designed the experiments, analyzed the data, prepared figures and/or tables, and approved the final draft.
- Yifeng Lin conceived and designed the experiments, analyzed the data, prepared figures and/or tables, and approved the final draft.
- Jiahuan Chen conceived and designed the experiments, authored or reviewed drafts of the article, and approved the final draft.
- Chenchen Bi conceived and designed the experiments, authored or reviewed drafts of the article, and approved the final draft.
- Zheng Liu conceived and designed the experiments, authored or reviewed drafts of the article, and approved the final draft.

## Data Deposition

This article is a literature review and did not utilize raw data.

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
