# Peer review of "Progress in the mechanism of autophagy and traditional Chinese medicine herb involved in alcohol-related liver disease"

_PeerJ, doi:10.7717/peerj.15977_

## Round 0.1 · original submission · Major Revisions

We have critical remarks on this work. Please extend the Discussion part on possible mechanisms involved in alcohol-related liver disease. Please check English again.

Reviewer 1 ·

Basic reporting

The idea of autophagy in ALD is relative new in recent years. This manuscript provides perspectives about Chinese herb medicie in the role of autophagy to treating ALD. I appreciate their works.

Experimental design

The search method of this manuscript is adequate. The tables and figures are comprehensive.

Validity of the findings

Same as methods, this review provide comprehensive works regarding the rols of Chinere herb medicine in the role of autophagy to treating ALD.

Additional comments

There are several minor points the authors may need to improve:

1. line 40. I suggest use alcohol "misuse" instead of "abuse"
2. line 62-66. These descriptions seems not so straightforward and hard to
understand. Please rephrase them.
3. line 80. Figure1. "Beclin-1" not "Becline-1"
4. line 92. The content of Figure1 seems not correspond with them. For example,
I can not find the "STX17" in Figure1?
5. line 127-133. There are some repititve descriptions. Please rephrase them.
6. line 134-154. Mitophagy section. I suggest the authors provide a figure to
depict these complex process for better understanding.
7. line 182. Please define what is the "DAFLD"
8. line 204. Please define what is the "TFEB"
9. line 296. I think the authors mean "apparent" mineralocorticoid excess
because their abbreviation "AME"
10. profressional english editing of this manuscript may be considered.

Reviewer 2 ·

Basic reporting

This literature review discussed the role of ingredients from several traditional Chinese medicine herb and the possible mechanisms involved in alcohol-related liver disease.

It is an interesting article and the content regarding the role of traditional Chinese medicine and autophagy in ALD has not been recently reviewed.

The authors provided to much information on autophagy in the beginning while the reason why the authors chose to discuss the traditional Chinese medicines was not mentioned.

Same sentences were repeatably used in the article. Some sentences are confusing.

Experimental design

Please include web address for all search engines used for the literature search. Several citations in the review were not paraphrased well, making it hard to understand. Please double check and re-summarize with more details.

Validity of the findings

The authors have cited numerous references in the manuscript. However, the whole paper lack the discussion about research findings and unresolved questions. It would also be helpful if the authors can provide structure image for each of the herbal extracts discussed. In addition, in table 1, the authors should add information about experimental model and treatment dosage for the cited studies.

Additional comments

Extensive editing of English language is suggested. Please also read over carefully to check for grammatical errors and consistency.

---

## Round 0.2 · accepted · Accept

Finally, both reviewers have no more remarks.

Reviewer 1 ·

Basic reporting

The authors responsed my questions clearly. I have no further comments.

Experimental design

The authors responsed my questions clearly. I have no further comments.

Validity of the findings

The authors responsed my questions clearly. I have no further comments.

Reviewer 2 ·

Basic reporting

The authors have addressed all the questions by the reviewer.

Experimental design

The authors have made changes to improve the review.

Validity of the findings

More information is added in the revised manuscript to improve its quality and validity.